# Bromatological Composition and In Vitro Ruminal Digestibility of Vaines of *Neltuma* spp. in Three Regions of the State of Zacatecas, México

**DOI:** 10.3390/vetsci12020142

**Published:** 2025-02-08

**Authors:** Eduardo Valdez-Romero, Lucía Delgadillo-Ruiz, Rómulo Bañuelos-Valenzuela, Eladio Delgadillo-Ruiz, Carlos Meza-López, Marisa Mercado-Reyes, Leticia Adriana Ramírez-Hernández, Francisco G. Echavarría-Chairez, Luz Adriana Arias-Hernández, Benjamín Valladares-Carranza, Rodrigo Flores-Garivay, Héctor Emmanuel Valtierra-Marín

**Affiliations:** 1Unidad Académica de Ciencias Biológicas, Universidad Autónoma de Zacatecas, Zacatecas CP 98068, Mexico; evaldez@uaz.edu.mx (E.V.-R.); marisa.mercado@uaz.edu.mx (M.M.-R.); 2Unidad Académica de Medicina Veterinaria y Zootecnia, Universidad Autónoma de Zacatecas, Zacatecas CP 98500, Mexico; luisbanuelos257@gmail.com (R.B.-V.); carmezlop@yahoo.com.mx (C.M.-L.); fechava1@yahoo.com (F.G.E.-C.); 3División de Ingenierías, Departamento de Ingeniería Civil y Ambiental, Universidad de Guanajuato, Guanajuato CP 36000, Mexico; e.delgadillo@ugto.mx (E.D.-R.); arhadriana@ugto.mx (L.A.A.-H.); 4Unidad Académica de Matemáticas, Universidad Autónoma de Zacatecas, Zacatecas CP 98600, Mexico; lramirez@uaz.edu.mx; 5INIFAP, Campo Experimental Zacatecas, Zacatecas CP 98500, Mexico; 6Centro de Investigación y Estudios Avanzados en Salud Animal, Universidad Autónoma del Estado de México, Facultad de Medicina Veterinaria y Zootecnia, Estado de Mexico CP 50295, Mexico; benvac2014@gmail.com; 7Instituto de Ciencias Agrícolas, Universidad Autónoma de Baja California, Mexicali CP 21705, Mexico; rodrigo.flores.garivay@uabc.edu.mx; 8Unidad Académica de Agronomía, Universidad Autónoma de Zacatecas, Zacatecas CP 98170, Mexico; hectorv@uaz.edu.mx

**Keywords:** *Neltuma*, quality, metabolizable energy, sheep, fermentation

## Abstract

*Neltuma* spp. species are an important resource in arid and semi-arid ecosystems of the country and serve, among other purposes, as an economic source of feed for livestock. Analyzing the bromatological composition and in vitro fermentation of *Neltuma* spp. pods in three regions of the state of Zacatecas, Mexico, as an alternative feed for sheep is an important activity. At the end of the research it was concluded that *Neltuma* spp. pods have acceptable levels of protein, carbohydrates and fiber.

## 1. Introduction

The state of Zacatecas is located in the north-central portion of Mexico. Due to its geographic location, as well as its physiographic and climatic characteristics, it exhibits a predominantly arid/semi-arid climate [1]. These regions are home to the mesquite tree species (*Neltuma* spp.), which is considered of both economic and ecological importance to northern Mexico. However, the lack of proper management of its utilization has led to a decline in its populations in this region [2].

The mesquite tree (*Neltuma laevigata*) was previously classified under the genus *Prosopis*, which historically included 44 recognized species [3]. However, this taxonomic classification has recently been revised. According to the study by Hughes et al. [4], evolutionary changes have resulted in the formation of distinct clades, leading to the creation of six separate genera: *Anonychium*, *Indopiptadenia*, *Prosopis*, *Strombocarpa*, *Xerocladia*, and *Neltuma*. All American species previously classified under *Prosopis* have now been reclassified under the genera *Neltuma* and *Strombocarpa*. For example, *Neltuma laevigata* is now part of the genus *Neltuma*.

*Neltuma laevigata* is an important biotic resource in the arid and semi-arid regions of Mexico. It serves as a source of food for both domestic livestock and wildlife, while its flowers produce pollen and nectar for honey production in apiculture [5]. The species is found in central and southern Mexico, specifically in the states of Aguascalientes, Durango, Guanajuato, Guerrero, Hidalgo, Jalisco, Michoacán, Morelos, Mexico State, Nuevo León, Oaxaca, Puebla, San Luis Potosí, and Zacatecas. Its distribution spans isohyets of 300 to 900 mm of annual rainfall and altitudes of up to 2300 m above sea level, with its most common occurrence between 1800 and 1900 m. It can withstand average maximum temperatures of up to 40 °C.

*Neltuma laevigata* appears as trees that can reach heights of up to 12 m, although they more commonly range between 6 and 7 m. It can also take the form of shrubs measuring 2 to 3 m in height. The fruit is a reddish pod, drupaceous in nature, elongated, straight, or arched, and measures 10 to 30 cm in length. When mature, it can be flat or cylindrical and contains between 12 and 20 seeds. The mesquite tree grows naturally in a variety of soils, preferably in flat, deep soils. According to the United States Department of Agriculture (USDA) soil classification, it thrives in entisols, xerosols, and molisols, which have a high-water retention capacity [6].

*Neltuma* species are a critical resource for the sustainability of arid and semi-arid ecosystems because they contribute to nitrogen fixation and soil conservation [7,8]. *Neltuma* trees hold ancestral value, as they have provided protein and energy-rich feed for humans and animals [9]. In Mexico, the most abundant species of the *Neltuma* genus is *N*. *laevigata*, which grows in ecosystems where annual rainfall ranges between 250 and 500 mm. This species is distributed across 40% of the national territory and thrives in soils unsuitable for agriculture [10].

The *Neltuma* spp. (mesquite) species can grow in regions with scarce rainfall, where it is particularly adapted to thrive. In northern Mexico, native forage resources, which constitute part of the normal flora, serve as an economical source of livestock feed. However, in these areas, native pastures are scarce, and supplementation is necessary during critical periods. Proper management of various plant species (including legumes and grasses) is essential to maximize their potential for animal production [11].

Studies on the nutritional contribution of *N*. *laevigata*, particularly its fruit (pod), suggest its potential as a sustainable source of high-quality raw material. García et al. [12] determined that up to 7% of a goat’s winter diet can consist of *Neltuma* pods. Armijo et al. [13] emphasized the importance of evaluating the chemical and nutritional composition of *Neltuma* pods based on geographic distribution and altitude, to identify their advantages and limitations as a feed source.

Ruiz et al. [14], Bravo et al. [15], and Choge et al. [16] reported that *Neltuma* spp. possesses a nutritional profile consisting of 7–22% protein, 30–75% carbohydrates, 11–35% crude fiber, 1–6% fat, and 3–6% ash. However, it is important to note that mesquite pods also contain antinutritional factors, including alkaloids, phenolic compounds, flavonoid glycosides, steroids, and tannins, which can have adverse effects on livestock. Consequently, their inclusion in animal diets must be carefully managed to mitigate the risk of toxicity [17].

*Neltuma laevigata*, as a species adapted to the arid and semi-arid conditions of Mexico, possesses the ability to grow, develop, and produce fruit under such challenging conditions. Its fruit contains the necessary nutrients to serve as an alternative or supplementary feed in the diet of sheep, particularly during drought periods when forage growth is inhibited, negatively affecting sheep production [18]. Therefore, the pods can be a usable resource for feeding ruminants, which will complement the maintenance requirements in the dry season. *Neltuma* spp. pods have the necessary nutrients to be used as alternative feed for sheep in times of drought without being affected by soil type or weather conditions.

The objective of the present study was to analyze the bromatological composition of *Neltuma* spp. pods and their in vitro fermentation in three regions of the state of Zacatecas, Mexico, as an alternative feed for sheep.

## 2. Materials and Methods

### 2.1. Study Site

Sampling was conducted in three climatologically and edaphologically contrasting regions (north, center, and south) of the state of Zacatecas, where *Neltuma* spp. grows. The study was carried out during May 2021 and 2022. The northern region, which includes the municipality of Mazapil, is characterized by a predominantly warm–dry and dry temperate climate, with an average annual temperature of 26.2 °C and annual precipitation of 388 mm. The soil in this region is predominantly clay-textured, with average values for the following characteristics: organic matter 3.84%, pH 8.66, NO_3_ (ppm) 14.76, P_2_O_5_ (ppm) 27.40, and K (ppm) 2.62.

The central region comprises the municipalities of Zacatecas, Guadalupe, and Jerez, which are characterized by predominantly dry temperate climates, with an average annual temperature of 22 °C and annual precipitation of 481 mm. The soil in this region is predominantly loamy-textured, with average values for the following characteristics: organic matter 1.08%, pH 7.89, NO_3_ (ppm) 13.71, P_2_O_5_ (ppm) 20.70, and K (ppm) 1.85.

The southern region, which includes the municipalities of Jalpa, Apozol, and Juchipila, is characterized by a predominantly warm-semidry climate, with a mean annual temperature of 29.9 °C and annual precipitation of 667 mm [19]. The soil in this region is also predominantly loamy-textured, with average values for the following characteristics: organic matter 1.99%, pH 8.36, NO_3_ (ppm) 16.12, P_2_O_5_ (ppm) 23.78, and K (ppm) 2.09.

The altitude of the sampling sites ranges from 1279 to 2358 m above sea level. Figure 1 shows the three regions sampled and the positions of the three sampling points.

According to the methodology described by Harden and Zolfaghari [20], sampling was carried out when the pods were in their immature stage. The pods were manually collected using pruning shears and polyethylene bags, aiming to fill each bag to an approximate weight of one kilogram. In total, 27 samples were collected from 3 sites in each of the 3 regions.

The *Neltuma* spp. pods were initially frozen using liquid nitrogen to halt the ripening process and preserve the samples. Subsequently, the pods were ground to a fine consistency for analysis.

### 2.2. Bromatological Composition Analysis

The chemical analysis included the determination of non-fibrous carbohydrates (NFC) (Method AOAC 985.29), ash (Cz) (Method AOAC 942.05), crude fiber (CF) (Method AOAC 2001.11), and crude protein (CP) using the micro-Kjeldahl method (Method AOAC 989.03) [21]. To determine the fibrous fraction, analyses of neutral detergent fiber (NDF), acid detergent fiber (ADF), and acid detergent lignin (ADL) were performed following the procedure described by Van Soest et al. [22].

Additionally, the evaluation of mineral components included total nitrogen, calcium, copper, iron, magnesium, manganese, zinc, sulfur, and potassium, which were determined using atomic absorption spectroscopy, while phosphorus was measured using visible spectrophotometry [23,24].

Based on the data obtained from the chemical analysis, digestible energy (DE), metabolizable energy (ME), net energy for lactation (NEL), net energy for maintenance (NEM), net energy for gain (NEG), and total digestible nutrients (TDN) were calculated according to NRC 2001 guidelines, as described by Salazar [24,25].

### 2.3. In Vitro Ruminal Fermentation

The in vitro gas production technique was conducted following the procedure described by Theodorou et al. [26]. Ground mesquite pods (*Neltuma laevigata*) were used as the substrate. To maintain an anaerobic environment, CO_2_ was employed and the incubation temperature was set at 39 °C. The ruminal inoculum was obtained from two male Rambouillet sheep, one year old (weighing 45 ± 3 kg), which had undergone a 30-day adaptation period on a diet consisting of 83% hay (alfalfa and wheat straw) and 17% concentrate (ground corn, soybean meal, macrominerals, and microminerals). The animals were slaughtered for human consumption at a municipal slaughterhouse in compliance with the Official Mexican Standard, NOM-033-SAG/ZOO-2014, which establishes the methods for the humane slaughter of domestic and wild animals.

Approval was obtained from the Bioethics and Animal Welfare Committee of the Faculty of Veterinary Medicine and Animal Sciences at the Autonomous University of Zacatecas (File 2023/04). The committee determined that the experiment did not require formal approval since no live animals were used, and based on the committee’s regulations, the experiment could proceed.

Alfalfa (*Medicago sativa* L.), the forage most commonly used in sheep diets in Mexico [27], was used as a control without the addition of any additives. The experiment was conducted over an 84 h period, with measurements taken at 3, 6, 9, 15, 27, 39, 51, 60, 72, and 84 h. Each sample was analyzed in triplicate. Fermentation units (FUs) (ZHANXUBIO, Boston Bottle, China) with a capacity of 120 mL were used for each sample. Gas production was measured using a Sper Scientific pressure meter Webster, TX, EE. UU., with cumulative gas pressure recorded in PSI.

The pH of the fermentation medium was measured at different digestion times using a Thermo Scientific Orion Star A211 Indonesia potentiometer. Ammonia concentration was determined using 50 µL of the sample, to which 25 mL of phenol reagent Merck KGaA, Darmstadt, Germany (A) and 2 mL of hypochlorite reagent Fisher Scientific S.L., Alcobendas, Spain (B) were added. The samples were boiled at 95 °C in a water bath for 5 min, cooled, and their concentrations were measured using a Jenway Genova Plus Stone, Staffs, UK spectrophotometer at 630 nm [28].

### 2.4. Statistical Analysis

The bromatological composition data were analyzed using ANOVA, a statistical model used to assess significant differences among three or more groups by partitioning total variance into between-group and within-group components [29,30]. To identify specific pairs of regions (north, center, and south) with significant differences, post hoc tests such as Tukey [31], Bonferroni, and Sidak were applied, with adjustments for multiple comparisons to control the type I error rate. For the fermentation analysis, Student’s *t*-test [32] was employed to compare means between two groups, accounting for data variability. Additionally, the Kruskal–Wallis test [33], a non-parametric method based on ranks, was used to evaluate differences across all regions when the assumption of normality was not met. Pairwise differences following the Kruskal–Wallis test were further assessed using the Dunn–Bonferroni test [34], which adjusts for multiple comparisons while maintaining robustness in non-parametric data. All statistical analyses were performed using the IBM SPSS statistical software package (version 29.0.20(20)), facilitating the accurate implementation of these methods.

## 3. Results

The results of the bromatological analysis of *Neltuma* spp. pods from the three regions of the state of Zacatecas are presented in Table 1. It can be observed that the dry matter content at all sampling points exceeds 94%. Regarding the crude protein content, samples from the central region exhibit the highest percentage, with an average of 20.86 ± 1.37%. For non-fibrous carbohydrate content, the highest average is 26.78 ± 1.32%, corresponding to the samples analyzed from the southern region.

The mineral content (Table 1) showed averages of calcium at 0.50%, phosphorus at 0.23%, magnesium at 0.15%, potassium at 1.72%, and sodium at 0.03%, with sodium being the mineral present in the lowest concentration. The ADF content ranged from 32.98% to 34.71%, while NDF content ranged from 41.77% to 43.53%.

The results indicate statistically significant differences among the regions for several key bromatological variables. Crude protein (CP) content was significantly higher in the central region (20.86 ± 1.37%) compared to the northern (18.30 ± 1.72%) and southern (18.18 ± 1.35%) regions (*p* < 0.05). For acid detergent fiber (ADF), the northern region exhibited the highest values (34.71 ± 2.52%), which were significantly greater than those in the central (33.22 ± 1.34%) and southern (32.98 ± 1.73%) regions (*p* < 0.05). Similarly, neutral detergent fiber (NDF) showed regional differences, with the northern region (43.53 ± 2.81%) presenting the highest values, followed by the southern (42.47 ± 1.26%) and central (41.77 ± 1.47%) regions (*p* < 0.05). Total digestible nutrients (TDN) were significantly higher in the central region (64.14 ± 1.21%) compared to the southern (62.78 ± 1.39%) and northern (61.33 ± 1.00%) regions (*p* < 0.05). These findings highlight the influence of regional variation on certain bromatological components, particularly protein and fiber fractions, which may affect the nutritional quality of *Neltuma* spp. pods across different regions.

Table 2 presents the analysis conducted using IBM SPSS software (version 29.0.20(20)), employing ANOVA with Tukey, Bonferroni, and Sidak post hoc tests to evaluate significant differences among regions (north, central, and south) across multiple bromatological and energetic variables. Significant differences were observed in the following parameters.

Crude Protein (CP): Both in dry matter (DM) and as fed (As Fed), with the central region (Region 2) exhibiting significantly higher concentrations (*p* < 0.05).

Lignin: Indicating variations in fiber composition across regions (*p* = 0.031).

ADICP: Highlighting differences in proteins bound to insoluble compounds (*p* = 0.029).

Total Digestible Nutrients (TDN): Regional differences indicate variability in energy content (*p* = 0.047).

Net Energy:For lactation (NEL) (*p* = 0.022).For maintenance (NEM) (*p* = 0.019).For gain (NEG) (*p* = 0.017).

These results suggest that certain regions, particularly the central region, have mesquite pods with higher energetic quality and nutrient content.

In addition, ammonia production data indicated similar behavior across regions, with values falling within ranges compatible with efficient microbial function. The pH remained above 6.0 in all regions, ensuring optimal bacterial activity and fermentation efficiency. However, differences in digestibility and pH were not statistically significant, suggesting that bromatological variations between regions do not substantially impact these parameters. These findings confirm the overall consistency of mesquite pods as a feed resource across regions.

Figure 2A displays the gas production curves (mL/g DM) for the three regions and the control group. The southern region exhibited the highest gas production (134.81 mL/g DM), indicating superior fermentation efficiency, while the northern region showed the lowest production (75.26 mL/g DM), potentially due to differences in bromatological composition. The ANOVA results confirmed significant differences (*p* < 0.05), aligning with the trends observed in this figure.

Figure 2B presents the results of the Dunn–Bonferroni test for gas production. Significant differences were observed between the regions and the control, with the southern region (labeled “c”) showing statistically significant differences (*p* < 0.001). The consistency between ANOVA and post hoc results further supports these findings.

In this study, Figure 2C shows the behavior of ruminal pH during fermentation. The average pH remained above 6.0 in all three regions, ensuring optimal conditions for bacterial activity and effective fiber digestion. The ANOVA results validate this stability across regions, highlighting the robustness of the observations.

Figure 2D compares the ruminal pH at the end of the 84 h fermentation process. All regions achieved pH values higher than the control, confirming the effective utilization of acid detergent fiber in *Neltuma* spp. pods. Statistical analyses reinforce that regional differences in pH were not significant, ensuring consistency in fermentation conditions.

The ammonia production observed in this research is presented in Figure 2E, which indicates behavior similar to the control, suggesting that the pods provide a suitable nitrogen source for microbial protein synthesis. The ANOVA results showed no significant differences between regions, supporting the uniformity in ammonia production observed in the figure.

Figure 2F displays the results of the Dunn–Bonferroni test for ammonia production. While slight variations were observed among the three regions, the statistical analysis confirmed that these differences are influenced by factors such as crude protein and fiber content.

Table 3 provides a comparison of the average performance at three selected time points for statistical analysis: the baseline, the midpoint of the process, and the endpoint. Each column in the table represents the average values along with their standard deviations. The values were compared using the Kruskal–Wallis statistical test, where results with *p*-values less than or equal to 0.05 were considered statistically significant, indicating that the observed effects are attributable to the process.

This table shows that the southern region consistently demonstrated the highest gas production among the experimental regions, with notable values at 3 h (20.8 ± 6.8 mL/g DM) and 84 h (134.8 ± 31.0 mL/g DM). In contrast, the control group consistently surpassed all experimental regions, achieving the highest gas production at 24 h (85.7 ± 0.1 mL/g DM) and 84 h (172.7 ± 9.5 mL/g DM), indicating superior overall efficiency. The higher variability observed in the southern region suggests potential heterogeneity in the samples, highlighting the need for further investigation into its bromatological characteristics.

## 4. Discussion

In this study, lignin values were 7.68% for the northern region, 6.60% for the central region, and 7.27% for the southern region. These values are comparable to those reported by Ángeles et al. [35], who found lignin contents ranging from 5.6% to 7.6% in the state of Hidalgo, Mexico. The acid detergent fiber (ADF) content ranged from 32.98% to 34.71%, while neutral detergent fiber (NDF) ranged from 41.77% to 43.53%. These values are higher than those reported by Ali et al. [36], who documented 7% ADF and 29.8% NDF.

During the in vitro fermentation process, lower gas production values were observed in the control (alfalfa) compared to the pod samples. De la Rosa et al. [37] and Reyes et al. [38] have reported the nutritional profile and bioactivity of mesquite flour derived from various *Neltuma* spp. species. These studies found that *Neltuma laevigata* pod flour has a good nutritional profile, including 10% protein, 3.6% fat, and 26.7% crude fiber.

In the present study, the crude protein content (20.86%) was higher than that reported by De la Rosa et al. [37] and other authors [13,39,40]. However, the carbohydrate content found in this study (26.78%) was lower than the 56.8% reported by De la Rosa et al. [37]. Similar findings were reported by Andrade et al. [41], who documented higher carbohydrate values (48.10% and 49.20%) compared to this research. The results obtained for non-fibrous carbohydrates in this study ranged from 24.94% to 26.78%, notably lower than the 48.1% and 49.20% reported by Rodríguez et al. [42] and García et al. [43]. This difference can be attributed to the stage of pod maturation, as sampling was performed at an early stage (60 days). Among the compounds present in the non-fibrous carbohydrate fraction are sugars, starches, organic acids, fructans, and pectins, which provide energy and thus represent an important source of energy in the diet of ruminants [44].

The TDN values obtained in this study ranged from 61.33% to 64.14%, higher than the 41.1% reported by Armijo et al. [8] and the 49% reported by García et al. [43], but lower than the 77.7% reported by Baraza et al. [45]. These results demonstrate significant variation in the energy value of this product.

Feed conversion efficiency is determined by fiber fermentation, which largely depends on the lignin content. Feeds with lower lignin content and higher cellulose and hemicellulose content exhibit greater efficiency [46]. According to Ramírez and Soto [47], lignin content has been linked to lower in vitro and in situ fermentation of dry matter (DM) and reduced volatile fatty acid concentrations in shrub leaves in Mexico. Cervantes et al. [48] reported a lignin content of 16.1% in *Neltuma glandulosa*.

In addition, ruminal pH and ammonia production during digestion must be considered. The ideal ruminal pH for microbial activity and proliferation ranges from 6.2 to 7.0. Within this range, fermentation processes are optimized, including the efficient breakdown of fibrous components in forage [49]. Ruminal pH varies depending on the type of feed, its form, and the frequency of administration. Diets high in non-structural carbohydrates tend to lower pH, while those rich in structural carbohydrates help maintain pH at the upper end of the optimal range [50].

Regarding neutral detergent fiber (NDF), Armijo et al. [13] reported values of 30.9%, while Mejía et al. [18] reported 34.7%. These findings are similar to those in this study, which reported NDF values of 34.71% in the northern region, 33.22% in the central region, and 32.98% in the southern region. Similar values were found in studies of tender pods (47.62%) and mature pods (27.74%), as reported by Montañez et al. [51].

Armijo et al. [13] reported that the dry matter (DM) fermentation of *Neltuma* spp. pods varies according to their maturity stage. For tender *Neltuma* spp. pods, the DM fermentation value was 66.99%, while for dry pods it was 64.81%. Barros et al. [52] reported higher values of 82.6% in mature pods of *Neltuma juliflora*. In the present study, the digestible DM values were 61.33% in the northern region, 64.14% in the central region, and 62.78% in the southern region. The nutritional value of mesquite pods varies significantly depending on their maturity stage, making this an important factor to consider, particularly when these pods are used as a primary ingredient or substitute in whole-grain diets [51].

Ángeles et al. [35] and Román [53] noted that carbohydrates in mesquite pods are located in the mesocarp, which complicates their extraction and fermentation. For this reason, pre-processing of the pods, such as by roasting or crushing, has been recommended before their use as animal feed [42,54]. In this study, sampling was conducted on pods at the immature stage. Ángeles et al. [36] also recommended roasting the pods to deactivate certain antinutritional components that may hinder the degradation of proteins and carbohydrates in mesquite seeds.

The duration for which ruminal pH remains below 6.0 is a critical factor limiting optimal dry matter (DM) digestion in the rumen [4]. A pH below 6.0 restricts the growth of cellulolytic bacteria on cellulose and cellobiose. In addition to directly reducing cellulase activity, low intracellular pH in these bacteria causes an increase in the pH gradient, leading to anion toxicity [55].

Proteolysis is one of the most critical processes in the rumen. Under normal conditions, the majority of dietary protein is degraded by the ruminal microbial biota, with ammonia produced as the end product. Ammonia serves as a substrate for microbial protein synthesis, resulting in most of the protein reaching the abomasum being of microbial origin [56].

Ángeles et al. [35] mentioned that the similarities observed in the in vitro fermentation patterns between collected pods could be associated with similar agroecological conditions and geographical proximity. Cerón et al. [57] pointed out that differences in fermentation patterns between municipalities are influenced by water and soil conditions, which have a direct impact on the content of biomolecules naturally produced through photosynthesis in plants. In this study, the sampling regions were geographically distant and exhibited different hydric and edaphological characteristics.

## 5. Conclusions

The statistical analysis supports and extends the findings of the article by providing clear evidence of regional variability in chemical composition and energetic parameters. The ANOVA results validate that those differences in crude protein (*p* < 0.05), lignin (*p* = 0.031), ADICP (*p* = 0.029), TDN (*p* = 0.047), and net energy (NEL, NEM, NEG; all *p* < 0.05) are statistically significant and not random. This highlights the central region’s higher protein content and the southern region’s greater gas production as key factors.

Despite these differences, the results indicate that the measured parameters of digestibility, ammonia production, and pH remain consistent across regions, reinforcing the suitability of mesquite pods as a reliable feed resource. These findings suggest that the bromatological variations between regions are not substantial enough to negatively affect ruminal digestion or fermentation efficiency. *Neltuma* spp. (mesquite) pods exhibit acceptable levels of protein, carbohydrates, and fiber across the three regions of the state. However, factors such as pod maturity, variety, climate, and soil characteristics can contribute to variations. Furthermore, the high content of nutrients, such as non-fibrous carbohydrates and crude protein, combined with the low neutral detergent fiber content in *Neltuma* spp. pods, suggests that they can be considered a viable alternative feed source for sheep.

## Figures and Tables

**Figure 1 vetsci-12-00142-f001:**
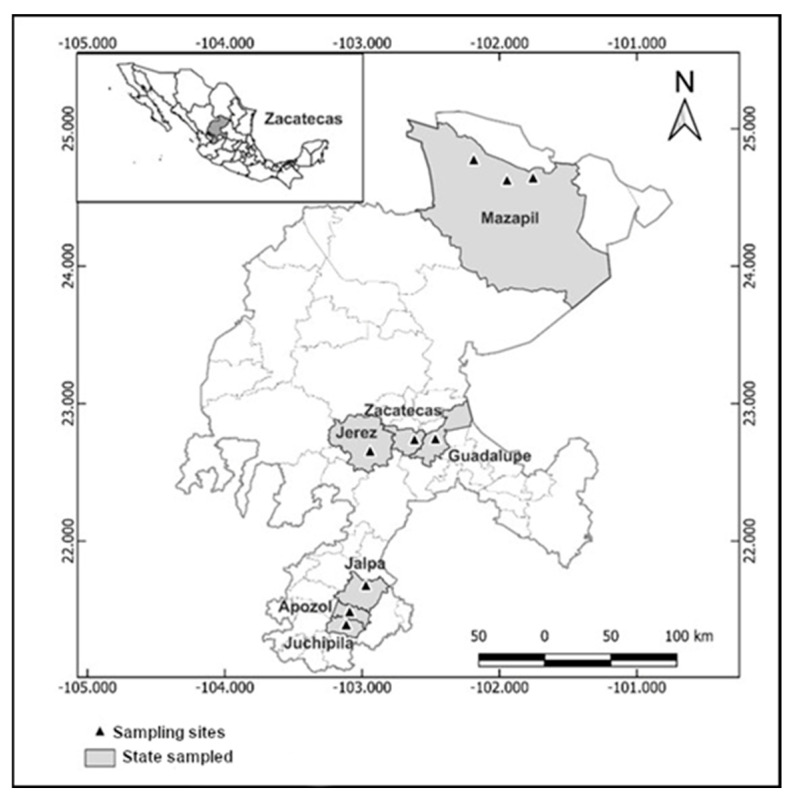
Distribution of sampling points in three regions of the state of Zacatecas.

**Figure 2 vetsci-12-00142-f002:**
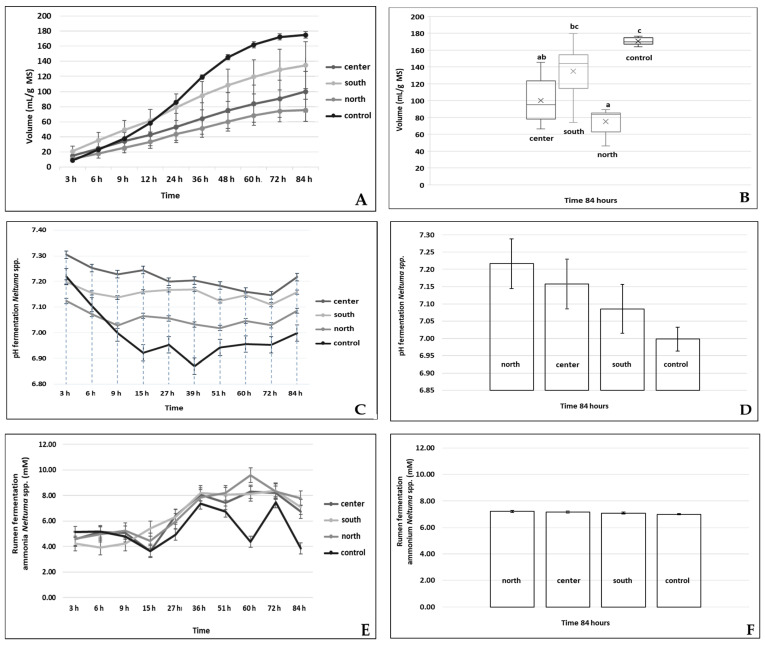
Results of the different statistical analyses. (**A**). Comparative gas production for the three regions and the control (bars indicate standard deviation). (**B**). Dunn–Bonferroni test analyzing differences in gas production between the three regions (ab: equal; bc: equal; a: different to b and c; c: different to all regions). (**C**). pH behavior during ruminal fermentation in the three regions. (**D**). Comparison of ruminal pH across the three regions using ANOVA. (**E**). Ammonia behavior during ruminal fermentation. (**F**). Dunn–Bonferroni test analyzing differences in ammonia production among the three regions (a: regions equal; b: different from all regions).

**Table 1 vetsci-12-00142-t001:** Bromatological analysis of *Neltuma* spp. pods in the northern, central, and southern regions of the state of Zacatecas.

Variables	Northern $\bar{x}$ ± DE	Central $\bar{x}$ ± DE	South $\bar{x}$ ± DE
Moisture, (%)	5.50 ± 0.36	5.87 ± 0.22	5.74 ± 0.25
Dry matter, (%)	94.50 ± 0.36	94.16 ± 0.25	94.27 ± 0.24
Crude protein, (%)	18.30 ± 1.72 ^a^	20.86 ± 1.37 ^b^	18.18 ± 1.35 ^a^
Available protein, (%)	17.60 ± 1.80	20.31 ± 1.33	17.48 ± 1.40
Insoluble acid detergent protein (% ADICP)	0.70 ± 0.12	0.54 ± 0.07	0.67 ± 0.10
Adjusted crude protein, (%)	18.30 ± 1.72	20.86 ± 1.37	18.18 ± 1.35
Soluble protein (% CP)	65.33 ± 5.00	68.15 ± 2.66	63.22 ± 4.71
Insoluble neutral detergent protein (% NDICP)	1.49 ± 0.40	1.28 ± 0.21	1.47 ± 0.10
Acid detergent fiber (% ADF)	34.71 ± 2.52 ^a^	33.22 ± 1.34 ^b^	32.98 ± 1.73 ^b^
Neutral detergent fiber (%NDF)	43.53 ± 2.81 ^a^	41.77 ± 1.47 ^b^	42.47 ± 1.26 ^c^
Lignin, (%)	7.68 ± 0.62	6.60 ± 0.22	7.27 ± 0.75
Non-fibrous carbohydrates (% NFC)	26.01 ± 3.37 ^a^	24.94 ± 2.40 ^a^	26.78 ± 1.32 ^a^
Starch, (%)	1.64 ± 1.00	1.76 ± 0.37	1.73 ± 0.70
Simple sugars (% ESC)	2.81 ± 0.68	3.71 ± 1.30	3.62 ± 0.39
Ash, (%)	4.78 ± 0.32	4.27 ± 0.28	4.66 ± 0.35
Calcium, (%)	0.50 ± 0.09	0.50 ± 0.06	0.46 ± 0.07
Phosphorus, (%)	0.23 ± 0.04	0.25 ± 0.02	0.23 ± 0.02
Magnesium, (%)	0.15 ± 0.02	0.16 ± 0.01	0.14 ± 0.02
Potassium, (%)	1.72 ± 0.19	1.75 ± 0.13	1.72 ± 0.12
Sodium, (%)	0.03 ± 0.00	0.03 ± 0.00	0.03 ± 0.01
Digestible energy, DE (Mcal/Kg)	2.84 ± 0.06	3.00 ± 0.03	2.90 ± 0.07
Metabolizable energy, ME (Mcal/Kg)	2.41 ± 0.06	2.58 ± 0.03	2.47 ± 0.07
Net energy lost, NEL (Mcal/Kg)	1.38 ± 0.04	1.49 ± 0.02	1.42 ± 0.04
Net maintenance energy, NEM (Mcal/Kg)	1.45 ± 0.05	1.58 ± 0.02	1.49 ± 0.05
Net energy gain, NEG (Mcal/Kg)	0.87 ± 0.04	0.98 ± 0.02	0.90 ± 0.04
Total digestible nutrients, TDN (%)	61.33 ± 1.00 ^a^	64.14 ± 1.21 ^c^	62.78 ± 1.39 ^b^

Notes: $\bar{x}$ = mean, SD = standard deviation, % = percentage, Mcal = megacalories, Kg = kilogram; mean values with different letters in the same row differ statistically (*p* < 0.001). The energetic variables DE, ME, NEL, TDN, NEM, NEG and TDN were calculated according to NRC 2001.

**Table 2 vetsci-12-00142-t002:** Results of ANOVA with Tukey, Bonferroni, and Sidak post hoc tests to evaluate significant differences among regions (north, central, and south).

Variables with Significant Differences									
Region	TDN (As Fed)	TDN (DM)	NEL (As Fed)	NEL (DM)	NEM (As Fed)	NEM (DM)	NEG (As Fed)	NEG (DM)	Zinc (DM)
North	58.0 b	61.1 b	1.32 b	1.39 b	1.21 c	1.28 c	0.67 b	0.71 c	43.0 b
Center	60.8 a	61.7 a	1.39 a	1.48 a	1.31 a	1.39 a	0.77 a	0.82 a	57.6 a
South	59.2 b	62.7 b	1.35 b	1.43 b	1.26 b	1.34 b	0.72 b	0.76 b	47.3 b

Letters: Indicate differences (*p* < 0.05) between regions.

**Table 3 vetsci-12-00142-t003:** Comparison of gas production times among the three regions (central, northern, and southern) and the control group using the Kruskal–Wallis test (*p* ≤ 0.05).

Gas Production Times	Central Region n = 9	Northern Region n = 9	Southern Region n = 9	Control n = 9	*p*
Time 3 h, average ± standard deviation	14.9 ± 4.8	11.1 ± 3.6	20.8 ± 6.8	8.6 ± 0.2	<0.001
Time 24 h, average ± standard deviation	53.1 ± 17.8	43.7 ± 11.5	79.0 ± 17.5	85.7 ± 0.1	<0.001
Time 84 h, average ± standard deviation	100.0 ± 26.9	75.2 ± 14.6	134.8 ± 31.0	172.7 ± 9.5	<0.001

The table demonstrates statistically significant differences in gas production among the central, northern, and southern regions and the control group at three measured time points (3, 24, and 84 h; *p* < 0.001). The control group consistently exhibited the highest gas production, while the southern region displayed the greatest variability in values.

## Data Availability

All datasets used in this study are readily accessible upon request to the authors.

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
