# Peer review of "Bromatological Composition and In Vitro Ruminal Digestibility of Vaines of *Neltuma* spp. in Three Regions of the State of Zacatecas, México"

_vetsci, 2025, doi:10.3390/vetsci12020142_

Round 1
Reviewer 1 Report
Comments and Suggestions for Authors
Dear Authors,
Your manuscript entitled "Bromatological composition and in vitro ruminsl digestibility of vaines of Neltuma spp in three regions of the state of Zacatecas, México" might be suitable for publication in Veterinary Sciences after major revision. Please, see below a list of comments/suggestions to be applied by you before accepting it.
Yours sincerely,
Reviewer.
L23 Write in italics "in vitro".
L24 Delete "as an alternative food".
L29 Write in italics "in vitro".
L37 Replace "recom-mended" by "reco-mmended".
L65 Replace "tan-nin" by "ta-nnin".
L66 Replace "live-stock" by "livestock".
L72 Replace "carbo-hydrates" by "carbo-hydrates".
L75-L80 Delete this paragraph by another one. Explain your main hypothesis and why it is relevant for research.
L82 Write in italics "in vitro" and Replace "Mex-ico" by " Me-xico".
L88 Replace "an-nual" by "a-nnual".
L90 Replace "pre-dominantly" by "predominantly".
L92 Replace "char-acterized" by "characterized".
L85-L97 Add information related to Neltuma and alfalfa varieties tested, climatic conditions and soil agronomic characteristics (pH, fertilisation, texture, etc.). Provide also details about the experimental design applied (treatments under study and repetitions).
L101-L104 Explain how pods were harvested (equipment used and maturity stage).
L120-L127 Add the protocol number approval got for running this experiment with animals. Specify the technique used for measuring in vitro ruminal digestibility. Provide set conditions for running in vitro trial (temperature, CO2, etc.).
L125 Replace "rec-orded" by "re-corded".
L135-L142 Describe the mathematical model used for comparison of means and explain the factors under study.
L136 Replace "co-efficient" by "coe-fficient".
L145 Replace "con-tent" by "content".
L144-L147 Comment significant variables from Table 1.
L147-L150 Delete from "For no" to "region".
L162-L180 Delete those paragraphs. Please, not repeat results provided in Table 2. Provide new information.
Table 2: Delete this table and perform a new one by considering the effect of the region (northern, center and south) in TDN, ME and DE using a comparison test for the means of those three variables.
L195-L213 Describe Figures 2A-F by order. Recheck the text to clarify your main ideas and to highlight them. Please, try to avoid any confusion and misinterpretation.
L203-L204 Replace "behav-ior" by "beha-vior".
L204-L205 Replace "val-ues" by "va-lues".
L226 Replace "col-umn" by "co-lumn".
Table 3: Add letters to show significant differences among treatments.
L235-L237 Delete this paragraph. No relevant information is provided.
L245 Write in italics "in vitro".
L249 Use scientific nomenclature to cite "Neltuma laevigata".
L257 Replace "var-iation" by "va-riation".
L262 Write in italics "in vitro" and "in situ".
L263 Use scientific nomenclature to cite "Neltuma glandulosa".
L273 Use scientific nomenclature to cite "Neltuma juliflora".
L294 Write in italics "in vitro".
L296 Replace "pat-terns' by "pa-tterns".
L303-L304 Justify "However, factors ... variations".
L309 Replace "Meth-odology" by "Metho-dology".
L324-L424 Check all References. Please, cite them according to Veterinary Sciences' instructions for authors.
Author Response
Good day
Dear reviewer
I sent article veterinary science -3414352
which was worked on to carry out the fulfillment of the observations indicated
I sent an attached file of how they were worked on
In the document they appear marked in red
I await any indication
Best regards
Dr. Lucía Delgadillo
Corresponding author

Reviewer 2 Report
Comments and Suggestions for Authors
This study is very innovative, but in the introduction, the research progress in this field is not fully elaborated, and the innovation of this study is not highlighted. Please ask the author to supplement some research background in this field, so that readers can better understand the significance and importance of this study.
Author Response
This study is very innovative, but in the introduction, the research progress in this field is not fully elaborated, and the innovation of this study is not highlighted. Please ask the author to supplement some research background in this field, so that readers can better understand the significance and importance of this study.
Reviewer 3 Report
Comments and Suggestions for Authors
This is a study of the characterization of the chemical composition and fermentation parameters (in vitro) of the Neltuma pod in different regions. Characterization studies are important and should be encouraged, however, the idea of comparing the composition of the marshes of three regions seems to me to be little explored in the study.
Introduction:
The authors need to make clear what motivated them to study the chemical characterization between the different regions of the state of Zacatecas. Are there consistent differences in rainfall or soil between these regions that could influence the composition? Are they different species (is there a previous study on this?)?
Paragraphs 68-80 seem out of place and should be removed. In vitro studies are nutritional assessment tools and should not be included in the introduction as a justification for the study. Please redirect this information to the discussion of the in vitro results.
Material and methods:
Lines 107-108. Specify the code for each method using the AOAC. Which reference was used for the non-fibrous carbohydrate equation? How and what was the reference for determining: starch, simple sugars, soluble protein, protein insoluble in neutral detergent and acid detergent? This information is essential for understanding the table.
Line 116-117. Provide the equations that were used to determine all variables (DE, ME, NEL, TDN...)
Result:
Table 1. Footer. Inform that the variables (DE, ME, NEL, TDN...) are calculated according to NRC (2001). By the way, why didn't you use the NASEM (2021) equations? Please do so. Adjust to the NASEM (2021) equations.
Figure 2. The images should be larger and more readable. The P values must be provided in the caption for all comparisons.
Table 3. Perform the Tukey test and add the different letters next to the means, within the table. This information is essential for understanding the table.
Discussion:
Articles cited in the discussion must have a DOI, please check this and adjust as necessary.
There is no discussion of the likely reasons for the differences observed between regions. Please check and adjust this.
Lines 278-284. Insert a characterization of the non-fibrous carbohydrates of Neltuma.
Lines 285-289. Insert the fractionation (CNCPS-Cornell) available in the literature for Neltuma. Lines 290-293. Insert the relationship of composition (crude protein, soluble protein, insoluble protein) with ammonia production in in vitro fermentation.
Conclusion:
Your data do not allow you to make the following inference “However, factors such as pod maturity, pod variety, climate, and soil characteristics can cause variations.” Remove and adjust your conclusion. I did not find a conclusion considering the differences between regions.
Author Response
Good day
Dear reviewer
I sent article veterinary science -3414352
which was worked on to carry out the fulfillment of the observations indicated
I sent an attached file of how they were worked on
In the document they appear marked in red
I await any indication
Best regards

Round 2
Reviewer 1 Report
Comments and Suggestions for Authors
Dear Authors,
Your manuscript might be now suitable for publication in Veterinary Sciences after including data in Table 2. Please, add the mean values of each parameter to it.
Yours sincerely,
Reviewer.
Author Response
In Table 2, the average values of each parameter were added as requested.
A general reading was carried out to verify the content, and all changes were marked in red.
Reviewer 3 Report
Comments and Suggestions for Authors
The authors made the suggested modifications. The manuscript has improved in quality and can be published.
Author Response
We appreciate your contribution with your review